# Overview of High-Dynamic-Range Image Quality Assessment

**DOI:** 10.3390/jimaging10100243

**Published:** 2024-09-27

**Authors:** Yue Liu, Yu Tian, Shiqi Wang, Xinfeng Zhang, Sam Kwong

**Affiliations:** 1Department of Computer Science, City University of Hong Kong, Hong Kong, China; yliu724-c@my.cityu.edu.hk (Y.L.); ytian73-c@my.cityu.edu.hk (Y.T.); shiqwang@cityu.edu.hk (S.W.); 2School of Computer Science and Technology, University of Chinese Academic of Sciences, Beijing 100190, China; xfzhang@ucas.ac.cn; 3School of Data Sciences, Lingnan University, Hong Kong, China

**Keywords:** high dynamic range, objective image quality assessment, subjective image quality assessment

## Abstract

In recent years, the High-Dynamic-Range (HDR) image has gained widespread popularity across various domains, such as the security, multimedia, and biomedical fields, owing to its ability to deliver an authentic visual experience. However, the extensive dynamic range and rich detail in HDR images present challenges in assessing their quality. Therefore, current efforts involve constructing subjective databases and proposing objective quality assessment metrics to achieve an efficient HDR Image Quality Assessment (IQA). Recognizing the absence of a systematic overview of these approaches, this paper provides a comprehensive survey of both subjective and objective HDR IQA methods. Specifically, we review 7 subjective HDR IQA databases and 12 objective HDR IQA metrics. In addition, we conduct a statistical analysis of 9 IQA algorithms, incorporating 3 perceptual mapping functions. Our findings highlight two main areas for improvement. Firstly, the size and diversity of HDR IQA subjective databases should be significantly increased, encompassing a broader range of distortion types. Secondly, objective quality assessment algorithms need to identify more generalizable perceptual mapping approaches and feature extraction methods to enhance their robustness and applicability. Furthermore, this paper aims to serve as a valuable resource for researchers by discussing the limitations of current methodologies and potential research directions in the future.

## 1. Introduction

Compared to ordinary Low-Dynamic-Range (LDR) images, High-Dynamic-Range (HDR) images exhibit a wider dynamic range (i.e., a larger ratio between the highest and lowest value in the image), resulting in superior overall contrast effects and richer detail, especially in the extremely bright or dark regions. Owing to their ability to offer highly realistic visual experiences to viewers, recent years have witnessed a significant increase in the research of HDR images, including photography [1], automotive [2], medical imaging [3], and remote sensing [4]. In this context, the HDR Image Quality Assessment (IQA) becomes a fundamental topic in the field of HDR image processing tasks, which can be widely employed for optimizing parameters in various codecs and learning-based models. However, it is not suitable to directly apply LDR IQA metrics to HDR images, due to the substantial disparities in statistical distribution and luminance range between LDR and HDR images. Therefore, specialized evaluation methods and metrics tailored for HDR images are essential to assess their quality effectively.

In general, existing IQA algorithms can be classified into subjective and objective methods according to the evaluation approach. With the explosive growth of HDR content, the pressing issue at hand pertains to the storage and transmission of HDR images. Consequently, existing subjective experiments [5,6,7] are dedicated to exploring the impact of compression algorithms on the visual quality of HDR images. Nevertheless, the subjective experiments are typically time-consuming and expensive. In contrast, objective methods are considered more efficient due to their ability to directly extract features that closely align with the human visual system (HVS) from the images. Specifically, the HDR-VDP family [8,9,10] models the luminance perceived by HVS. Some methods evaluate the quality of HDR images in the perceptual domain [11,12], while others adapt existing metrics to predict quality scores [13,14]. Despite great progress, HDR IQA still encounters some challenges, particularly in the lack of large subjective databases and the difficulty of properly compressing the dynamic range of the HDR image into the perceptual domain.

The necessity for a comprehensive survey of HDR IQA methods arises from the limitations of existing surveys, which fail to cover the scope of the field comprehensively or lack summaries of the latest methodologies [15]. Therefore, this paper aims to fill this gap by providing an extensive overview of existing HDR IQA algorithms, including both subjective and objective methods. Specifically, our statistical evaluation, which includes nine LDR IQA metrics and three perceptual mapping methods, demonstrates that the choice of mapping method and dataset characteristics substantially influence performance outcomes. Furthermore, we discuss the limitations of current subjective and objective HDR IQA approaches and potential directions for improvement, hoping to provide comprehensive insight for future research.

The structure of this paper is outlined as follows: Section 2 introduces existing subjective HDR IQA databases, including detailed descriptions of the database and experimental settings. Section 3 summarizes existing objective HDR IQA algorithms. The limitations of both subjective and objective HDR IQA approaches are discussed in Section 4, together with the potential directions for future research.

## 2. Subjective HDR Image Quality Assessment Databases

Existing subjective HDR IQA databases focus on the evaluation of the visual quality of HDR images with compression distortions. Therefore, we first introduce the existing HDR image compression schemes in Section 2.1 to provide some basic information that can help to understand the construction and experimental settings in the subjective databases. Subsequently, the details of the databases are described in Section 2.2.

### 2.1. Existing HDR Image Compression Scheme

In contrast to LDR images that are typically encoded in 8-bit integers, the most commonly used formats for HDR images are Radiance RGBE and OpenEXR, where the data are stored in 16-bit or 32-bit floating-point numbers. Therefore, traditional image compression schemes like JPEG and JPEG2000, designed for LDR images, can not handle the compression of HDR images directly. Due to the widespread application of LDR codecs, converting the HDR image to a LDR image to make it compatible with existing codecs is a highly intuitive approach with practical applications. As shown in Figure 1, the existing compression scheme [16,17] for HDR images can be summarized into three steps, detailed as follows: In the pre-processing stage, the input HDR image is converted to an LDR image by a Tone-Mapping Operator (TMO). Herein, Sugiyama et al. propose to improve the quality of the tone-mapped LDR image by an optimization module [17]. Afterward, the LDR image is fed into the codec (JPEG or JPEG2000), together with the additional information such as the parameters for inverse TMO (iTMO), or the different information between the HDR and corresponding LDR image. In the post-preprocessing stage, the content to be displayed varies according to the conditions of the terminal monitor. If the monitor supports HDR content, the HDR image will be generated from the decoded LDR image by the iTMO with additional information. Otherwise, the decoded LDR image can be directly displayed. Based on the previous research works on HDR image compression with JPEG and JPEG2000, a new, dedicated compression standard, designed for HDR images, is proposed by the JPEG committee, called JPEG-XT [18], where the LDR image is compressed in the base layer, and the overall dynamic range is provided in the extension layer. The latest video coding scheme, Versatile Video Coding (VVC) [19], also supports the compression of the HDR content, where the luma mapping with chroma scaling and luma-adaptive deblocking [20] are proposed to improve the performance of the visual quality of the compressed HDR content.

With such architecture, the visual quality of the compressed HDR images has been evaluated in several subjective databases. Narwaria et al. built two databases to evaluate the influence of the optimization methods [5] in the pre-processing stage and the effectiveness of different TMOs [21] under the JPEG and JPEG2000 scheme, respectively. Korsunov et al. [6] compressed the HDR images using the JPEG-XT scheme. Valenzise et al. [7] constructed a new database including the JPEG, JPEG2000, and JPEG-XT schemes to evaluate the performance of the objective HDR IQA algorithms. Afterward, Zerman et al. [22] extended the content in [7] and add a new TMO method, Perceptual Quantization (PQ) [23], in the proposed database. Recently, Mikhailiuk et al. [24] constructed a large HDR IQA database by combining two existing HDR IQA databases [5,6], where the MOS values are aligned by the psychometric scaling method [25]. Liu et al. [26] first proposed a large database to evaluate the performance of various HDR IQA metrics on JPEG-XT and the newest compression scheme VVC. More details of the databases are summarized in Table 1 and described as follows.

### 2.2. Existing HDR Databases for Quality Assessment

To evaluate the performance of (1) optimization methods in HDR compression and (2) the objective IQA algorithms for compressed HDR images, Narwaria et al. generated a database [5] and conducted a subjective experiment to obtain the corresponding mean opinion score (MOS). The database is composed of 10 reference HDR images and 140 distorted HDR images, which are compressed with the JPEG scheme with seven bitrates and two optimization methods. The Mean Square Error (MSE) and Structural Similarity Index (SSIM) [27] are used to optimize the quality of the generated LDR image with the iCAM06 [28] TMO. The quality scores of the 150 HDR images are collected from 27 observers using the absolute category rating with hidden reference (ACR-HR) methodology [29]. The subjective experiment is conducted in a room with 130 cd/m^2^ luminance condition. The HDR images are displayed on a SIM2 Solar47 HDR monitor [30] with a maximum luminance value of up to 4000 cd/m^2^. From the experimental results, the SSIM does not outperform MSE when they are used to optimize the quality of tone-mapped LDR images. Furthermore, the HDR-VDP-2 [9] shows better performance on the overall results. However, when it comes to statistical analysis, the HDR-VDP-2 is indistinguishable from the SIQM method [31].

**Table 1 jimaging-10-00243-t001:** An overview of existing HDR IQA databases. The observers, reference images, distorted images, distortion type, and subjective methodology are denoted as Obs., Ref., Dis., Dist., and Meth., respectively.

Database	#Obs.	#Ref.	#Dis.	Dist.	TMO	Meth.
Narwaria2013 [5]	27	10	140	JPEG	iCAM06	ACR-HR
Narwaria2014 [21]	29	6	210	JPEG2000	AL, Dur RG, RL Log	ACR-HR
Korshunov2015 [6]	24	20	240	JPEG-XT	RG MT [32]	DSIS
Valenzise2014 [7]	15	5	50	JPEG JPEG2000 JPEG-XT	Mai	DSIS
Zerman2017 [22]	15	5	50	JPEG JPEG2000	Mai, PQ	DSIS
UPIQ [24] (HDR part)	20	30	380	JPEG JPEG-XT	iCAM06 RG MT [32]	-
HDRC [26]	20	80	400	JPEG-XT VVC	RG PQ	DSIS

In the database proposed by Narwaria et al. [21], the performance of five TMOs under the JPEG2000 compression scheme is subjectively evaluated. Herein, three local TMOs proposed by Reinhard (RL) [33], Durand (Dur) [34], and Ashikmin (AL) [35], together with two global TMOs, logarithmic TMO (Log) and the global TMO (RG), proposed in [33], are applied in the experiments. In this database, six reference HDR scenes are selected to obtain 210 distorted HDR images with 7-bit rates for each TMO. The subjective experiment settings are consistent with the conditions in [5]. The MOS values of the 216 HDR images are obtained from 29 participants with ACR-HR methodology. The experimental analysis results reveal that the AL TMO statistically outperforms the other TMOs. Furthermore, the local TMOs can help to achieve better visual quality of the compressed HDR images, while the performance of the two global TMOs does not show statistical differences.

Korsunov et al. [6] compressed the HDR image with the JPEG-XT [36] compression scheme and conducted a subjective experiment to build an HDR IQA database. In this database, 20 reference HDR images with various content are cropped and adjusted to the size of 944×1080 for display. A total of 240 compressed HDR images are generated by applying three profiles and 4-bit rates for each scene. The subjective experiment is conducted in a room with 20 cd/m^2^ background luminance value. A SIM2 HDR monitor with luminance values varying from 0.001 to 4000 cd/m^2^ is adopted to display the HDR images. The double-stimulus impairment scale (DSIS) methodology [37] is applied to acquire the MOS values from 24 subjects.

Valenzise et al. [7] have developed a database specifically for evaluating full-reference HDR IQA metrics. The database is composed of 5 reference HDR images and 50 distorted HDR images, compressed using the JPEG, JPEG2000, and JPEG XT schemes. The TMO used in this database is proposed by Mai et al. [38]. The subjective experiment is conducted in a room with a background luminance value of 20 cd/m^2^. The HDR images are presented on a SIM2 HDR47 monitor. The MOS scores are collected by the DSIS methodology from 15 subjects, where the continuous quality scores from 0 to 100 are divided into five scales, denoting the distortion level from “very annoying" to “Imperceptible”. Based on this database, they evaluate the performance of the objective IQA methods, including HDR-VDP-2 [9] tailored for HDR images, alongside PSNR and SSIM, which were adapted for the assessment of perceptual mapped HDR images by logarithmic function or perceptual uniform (PU) encoding [39]. The results show that PU-SSIM outperforms other metrics, followed by log-SSIM and HDR-VDP-2.

In order to conduct a more comprehensive assessment of the performance of full-reference HDR IQA metrics, Zerman et al. [22] introduced a novel database, building upon their previous work [7]. This database comprises 5 reference HDR images and 50 HDR images distorted by the JPEG and JPEG2000 compression schemes. Herein, two TMOs are employed to generate the LDR images: the TMO proposed by Mai et al. [32], and the Perceptual Quantizer (PQ) [23] published in SMPTE ST 2084. The subjective experiment settings are controlled to be the same as in Valenzise2014 [7]. Subsequently, the obtained MOS values are further aligned with the MOS values in [7] for further analysis. In the experiment, 25 IQA metrics are evaluated on the proposed database and four existing databases. There are three interesting conclusions that can be drawn from the results. Firstly, the Narwaria2014 [21] is a much more challenging database due to its complex distortion types. Secondly, metrics tailored for HDR content, such as HDR-VDP-2.2 [40] and HDR-VQM [11], exhibit superior performance in comparison to other IQA metrics. Nevertheless, the probability of misclassification still remains as high as 20%, indicating the imperative for the design of HDR IQA metrics that better align with HVS. Furthermore, concerning compression distortions, structural loss in the luminance domain is more perceptible to the HVS than loss in the color space.

Due to the limited number of images in the previous HDR IQA databases, Mikhailiuk et al. [24] constructed a Unified Photometric Image Quality (UPIQ) database with 380 HDR images and 3779 LDR images. The HDR images are collected from the two databases Korsunov2015 [6] and Narwaria2013 [5]. The MOS values are aligned by the psychometric scaling methodology, where the rating and ranking data are combined to capture the true underlying quality of images. Specifically, the rating scores, such as MOS, are linearly mapped to a psychometric scale, while pairwise comparisons are analyzed to determine the probability of one image being rated higher than another. The unified scale was achieved by performing additional subjective cross-dataset comparisons, which were then scaled using Thurstone’s model [41] to ensure that all datasets adhered to a common quality scale. This method enhances precision and comparability across datasets that use different scoring methods, ensuring that scores from different experimental protocols can be directly compared.

Recently, Liu et al. [26] proposed an HDR compression (HRDC) database, with the goal of examining the perceptual quality of HDR images compressed using the JPEG-XT and VVC codecs. The experiment is conducted using a Samsung QN90A TV, with 20 participants providing opinion scores based on the DSIS testing methodology. This database is the largest open-source database available for the HDR IQA task, including 400 distorted HDR images with JPEG-XT and VVC compression and 80 reference HDR images. Based on this database, they conduct extensive experiments to evaluate the effectiveness of both HDR and LDR IQA metrics on the HDR IQA task. Furthermore, they also evaluate the effects of different perceptual mapping methods, such as PQ, PU, and PU21 mapping methods, on the performance of LDR IQA metrics. Based on the experimental results, these perceptual mapping methods exhibit significant variations in performance across different databases. Moreover, no single perceptual mapping method achieves optimal performance across all databases, underscoring the challenges posed by HDR image evaluation and suggesting the need for new mapping methods to achieve stable and effective quality assessment for HDR images.

## 3. Objective HDR Image Quality Assessment Methods

Recent years have witnessed significant developments of efficient objective LDR IQA metrics [42,43,44,45,46,47,48,49], which can serve as crucial constraints for various high-level tasks [50,51]. Since both HDR and LDR images are 2D visual data, can we directly use LDR IQA metrics to evaluate HDR image quality? To answer this, it is crucial to first identify the differences between HDR and LDR images. In HDR images, pixel values are linearly related to the natural luminance. On the contrary, during the capture process of LDR images (as shown in Figure 2), the camera sensors first involve a clipping operation to restrict the maximum value of the image. Consequently, the nonlinear mapping process compresses the dynamic range of the image using the Camera Response Functions (CRFs). Finally, the quantization step aims to alleviate the storage burden on cameras. In Figure 3, we provide a more intuitive way to illustrate the statistical differences between the HDR and LDR images. It is evident that the HDR image has a wider dynamic range. Furthermore, there are over-exposed regions where many pixel values are 255 in the LDR image. However, there are no such regions in the HDR image. Therefore, directly applying IQA metrics specifically designed for LDR images to assess the quality of HDR images is unreasonable due to the different data distributions between HDR and LDR images. In Zerman et al. [22] and Liu et al. [26]’s works, extensive experimental results also demonstrate a significant decline in the performance of the LDR IQA algorithms when they are directly applied to HDR images without any perceptual mapping functions.

Consequently, the effective evaluation of the quality of the HDR images remains a challenging task. Existing HDR IQA methods can be roughly divided into the following three categories according to the design strategies: (1) HVS-modeling-based HDR IQA methods: directly evaluating the image quality by modeling how the human eye perceives and processes the light signals from natural scenes; (2) existing IQA-metrics-based HDR IQA methods: such algorithms aim to make full use of existing HDR or LDR IQA metrics to effectively assess the quality of HDR images; (3) PU-encoding-based HDR IQA methods: assessing the quality of HDR images specifically within the PU domain. More details of the existing HDR IQA methods are summarized in Figure 4, Table 2 and described in Section 3.1 and Section 3.2, respectively.

### 3.1. HVS-Modeling-Based HDR IQA Methods

The HDR-VDP family comprises a widely used and efficient set of HDR IQA algorithms, where the first version, HDR-VDP-1 [8], is proposed by Mantiuk et al., followed by the improved HDR-VDP-2 [9], HDR-VDP-2.2 [40], and the latest HDR-VDP-3 [10]. The basic idea of the HDR-VDP family is to model the HVS. Building on the Visual Difference Predictor (VDP) [53], HDR-VDP-1 first models the HVS within three stages: (1) modeling the light scattering using the Optical Transfer Function (OTF) [54]; (2) modeling the nonlinear response of the photoreceptors by the Just-Noticeable Difference (JND); (3) removing the less-sensitive frequency according to the Contrast-Sensitive Function (CSF). Subsequently, the extracted visual features are decomposed and then fed into a pooling module to calculate the visual difference map. Herein, no single quality score is provided in HDR-VDP-1. In HDR-VDP-2, the multi-band pooling strategy is proposed to obtain the quality score of the distorted HDR image. Additionally, photoreceptor response is modeled using sensitivity functions [55] rather than JND. In the pooling stage, the weighting parameters are optimized using the LIVE database [56] due to the lack of HDR IQA databases. Narwaria et al. improved the performance of HDR-VDP-2 by optimizing the parameters using two HDR databases [5,21] in HDR-VDP-2.2. Recently, the HDR-VDP-3 metric has been proposed, where the impact of different ages on visual quality is considered, together with the modeling of the local luminance. Additionally, the UPIQ database is used for parameter calibration to enable the quality evaluation of both HDR and LDR images. Despite significant success in HDR IQA, the HDR-VDP family still presents aspects worthy of improvement. Firstly, the parameters involved in simulating the HVS and the pooling stage necessitate intricate calibration processes [7]. Secondly, based on the experimental results in [5], HDR-VDP-2 appears to over-penalize the distortions for HDR images with higher quality.

Aydin et al. [57] proposed a Dynamic Range Independent Metric (DRIM) to measure the visual difference between two images with arbitrary dynamic range. Herein, they first model the HVS to extract the contrast features. Afterward, the cortex transform [58] is applied to obtain three distortion types separately, which are later used to generate the final visual difference map. Similar to HDR-VDP-1, this method tends to predict the probability of the difference between two images, yet it lacks the quantitative evaluation result, such as the MOS value. Given the same distorted image, Aydin et al. [39] found that the perceived distortion is more pronounced when viewed on an HDR display compared to an LDR display. Motivated by the above observation, Kottayil et al. [59] proposed a Deep Neural Network (DNN)-based blind IQA for HDR images. The network is composed of an E-Net for error prediction and a P-Net to emulate the perceptual resilience of the HVS, thereby quantifying the visibility of discrepancies within image blocks. The network is trained on the database that combines five HDR databases [5,6,7,21,22], where the MOS values are aligned by the Iterated Nested Least Square Algorithm (INLSA) [60]. Despite employing 5-fold cross-validation, the fact remains that only 552 distorted images are utilized for training in each fold, which is notably insufficient for a DNN model.

**Figure 4 jimaging-10-00243-f004:**
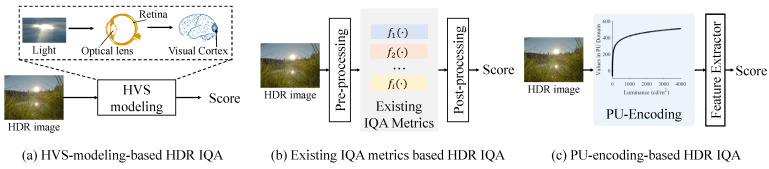
Overview of objective HDR IQA methods. The HDR images presented in this paper are tone-mapped for visualization by Gamma correction.

### 3.2. PU-Encoding-Based HDR IQA Methods

PU encoding, introduced by Aydin et al. [39], aims to convert HDR images into a perceptual domain, enabling the HDR image quality evaluation by LDR IQA metrics, which has been proven to be an effective mapping method [22]. Therefore, existing PU-encoding-based HDR IQA methods apply PU encoding first, followed by the quality evaluation in the PU domain. HDR video quality metric (HDR-VQM) [11] is the first HDR video quality assessment metric, and its efficacy in evaluating the quality of HDR images has been substantiated through subjective experiments [12,22,24]. In this metric, the PU encoding is first applied to obtain the perceived luminance map. Subsequently, the perceptual error is calculated by the log-Gabor [61] filter. The final quality score of the video is predicted by a spatial and temporal pooling strategy. The network proposed by Jia et al. [62], involved designing a Convolutional Neural Network (CNN) for blind HDR IQA. In the experiment, the network trained on PU-encoded HDR images shows significant improvement compared with the network trained on original HDR images. The PU-PieAPP proposed by Mikhailiuk et al. [24] combines the PU encoding with the PieAPP [63] for the quality evaluation of both HDR and LDR images. The network is trained on the UPIQ database [24], including 80 HDR images and 3779 LDR images. Liu et al. [12] introduce the local–global frequency feature-based model (LGFM). Herein, the odd log-Gabor filter is applied to obtain the local frequency feature, while the Butterworth filter is employed to simulate the CSF to generate the global frequency feature. The two feature maps undergo separate pooling processes and are subsequently integrated to yield the perceptual quality score.

**Table 2 jimaging-10-00243-t002:** Overview of existing HDR IQA methods.

Method	Year	Classification	Major Category
HDR-VDP-1 [8]	2005	Full-Reference	HVS-modeling-based HDR IQA methods
DRIM [57]	2008	Full-Reference	HVS-modeling-based HDR IQA methods
HDR-VDP-2 [9]	2011	Full-Reference	HVS-modeling-based HDR IQA methods
HDR-VDP-2.2 [40]	2015	Full-Reference	HVS-modeling-based HDR IQA methods
HDR-VQM [11]	2015	Full-Reference	PU-encoding-based HDR IQA methods
Kottayil et al. [59]	2017	No-Reference	HVS-modeling-based HDR IQA methods
Jia et al. [62]	2017	No-Reference	PU-encoding-based HDR IQA methods
HDR-CQM [13]	2018	Full-Reference	Existing IQA metrics based HDR IQA methods
PU-PieAPP [24]	2021	Full-Reference	PU-encoding-based HDR IQA methods
HDR-VDP-3 [10]	2023	Full-Reference	HVS-modeling-based HDR IQA methods
LGFM [12]	2023	Full-Reference	PU-encoding-based HDR IQA methods
Cao et al. [14]	2024	Full-Reference	Existing IQA-metrics-based HDR IQA methods

### 3.3. Existing IQA-Metrics-Based HDR IQA Methods

Currently, there is a limited number of HDR IQA algorithms, whereas there are numerous efficient LDR IQA methods. Therefore, it is common practice to directly employ existing LDR IQA methods for quality evaluation after mapping HDR images to the perceptual domain. In the early stages, Aydin et al. [39] introduced the PU encoding to convert HDR images into the perceptual domain. Recently, Mantiuk and Azimi proposed the PU21 [64] to model the new perceptual uniform space based on a new CSF [65]. Additionally, PQ [23] and Hybrid-Log–Gamma (HLG) [66] are two commonly used transfer functions in codecs, aiming to provide the optimal visualization results of the HDR content on the monitors. Note that PQ and HLG are not dedicated designed for HDR IQA, but they can also be adopted as the perceptual mapping functions. Although these mapping functions can effectively convert the HDR image into the perceptual domain, they are usually non-differentiable, making them unsuitable for the optimization of the learning-based model. Currently, there are four widely used differentiable mapping functions, including the logarithm function, Gamma correction, hyperbolic tangent function (tanh), and μ-law, which can be used to optimize the learning-based models [67,68,69]. To be specific, the logarithm function and Gamma correction model the nonlinear perception of light in the HVS, which are derived from the Weber–Fechner law and Stevens’ power law [70], respectively. Tanh is an extensively employed nonlinear function that can compress the dynamic range of the image into [−1,1]. The effectiveness of the μ-law algorithm has been demonstrated in various HDR-related tasks [71,72] for dynamic range compression, which is a differentiable companding algorithm originally designed for audio signals.

In addition, Choudhury and Daly proposed to evaluate the quality of HDR images by combining the results of various HDR IQA metrics (HDR-CQM) [13]. To be specific, they first selected six HDR IQA metrics with top performance according to the experimental results in [73]. Subsequently, the Sequential Floating Forward Selection (SFFS) [74] algorithm is applied to find the combination of metrics with the best performance. Afterward, the Support Vector Machine (SVM) is trained to predict the final quality score. Cao et al. [14] recently designed an inverse display model to convert the HDR image into the LDR domain to improve the performance of existing LDR metrics. Specifically, given an input HDR image, a set of LDR images with different exposures is generated by the proposed inverse display model. Afterwards, an LDR IQA metric is used to evaluate the quality of the image set. In addition, the LDR image stacks are also used to obtain the local exposure weights. The final quality score of the HDR image is obtained after the weighting pooling stage.

## 4. Discussion and Conclusions

This paper provides a comprehensive overview of existing subjective HDR IQA databases and objective HDR IQA metrics. While this research represents significant advancements, HDR IQA is still a challenging task, which is evolving alongside the generation of HDR content and the development of compression schemes. Therefore, this section further discusses the limitations of existing methods, along with the potential future work, aiming to provide a comprehensive understanding of the challenges and opportunities in the field of HDR IQA.

### 4.1. Subjective HDR IQA Databases

For the construction of HDR IQA subjective databases, we can observe three main limitations. *(1) There is a limited number of both reference and distorted HDR images in existing databases.* The content diversity of a database is crucial, as it not only affects the generalizability of the objective IQA algorithm evaluated on it but also impacts the robustness of the DNNs. The DNNs, owing to their powerful feature extraction capabilities, have been applied in numerous IQA models. However, this necessitates a large amount of training data for support. Unfortunately, the current HDR IQA databases are insufficient to train generalized deep learning-based HDR IQA models. *(2) The databases currently focus primarily on limited compression distortions.* As compression technologies evolve, commonly used compression techniques for HDR images have transitioned from JPEG and JPEG2000 to JPEG-XT and VVC [19]. Consequently, research on the effects of new compression distortion types on the human visual system remains insufficient. Moreover, there is a lack of research on other types of distortions, including motion distortions that may occur during the capture process, noise, and contrast distortions that may arise during the display process. *(3) All subjective experiments in existing HDR databases are conducted based on professional HDR monitors.* While these monitors offer high brightness levels, they are relatively costly and have limited application scopes. In contrast, currently widely used commercial HDR displays, although unable to achieve the brightness levels of professional monitors, still provide satisfactory visual experiences. Therefore, there is a need for subjective experiments on commercial displays to more accurately cater to the needs of a wider range of application scenarios. Based on the insights above, generating a larger database, exploring the effects of various distortion types, as well as conducting subjective experiments on commercial HDR displays can be potential research directions in the subjective HDR IQA database construction.

### 4.2. Objective HDR IQA Metrics

Despite the proliferation of HDR content, there are relatively few algorithms specifically designed for HDR image quality evaluation. Therefore, it is still a valuable undertaking to design robust, reliable, and perceptually aligned mapping methods that can map the brightness and color of the natural world to the perceptual domain that can be better aligned with HVS. *For the HVS-modeling-based approaches, the current challenge lies in effectively modeling the HVS*. HVS modeling presents significant complexity due to the large number of parameters involved and its reliance on experimental data, which are influenced by numerous factors. For instance, recent updates to the CSF by Ashraf et al. [75] highlight the importance of color, luminance, area, spatiotemporal frequency, and eccentricity as critical components of CSF modeling, which is an important part when modeling HVS. Such factors introduce considerable variability, as they are sensitive to both individual differences and environmental conditions. The fundamental issue lies in how to unify these multiple factors into a cohesive framework that can simplify the evaluation process while maintaining the accuracy of HVS-based predictions. *For the PU-encoding-based and existing-IQA-metrics-based approaches, there are mainly two factors that influence the effectiveness of the metrics, perceptual mapping functions, and the feature extraction method.* Herein, we conducted a comprehensive statistical analysis using the left-tail F-test, as suggested in [76], across five publicly available databases, to evaluate the performance of three perceptual mapping methods (Gamma, PQ, PU21) combined with nine LDR IQA metrics (PSNR, SSIM [27], MS-SSIM [77], FSIM [78], GMSD [76], VIF [79], ESIM [80], GFM [81], and DISTS [43]). As shown in Figure 5, cells highlighted in green (with a value of 1) indicate that the model corresponding to the row significantly outperforms the model corresponding to the column, while a value of 0 indicates no significant difference. From these results, we can draw the following key observations: (1) The choice of perceptual mapping method affects the accuracy of IQA model evaluations. When different mapping methods are applied to these IQA metrics, the performance ranking of IQA models can change. For instance, on the HDRC database, FSIM performs better than GFM when using the Gamma mapping, but there is no significant difference when using PQ or PU21. This indicates that perceptual mapping methods have a considerable impact on how well IQA models align with subjective quality assessments, highlighting the importance of selecting an appropriate mapping strategy for accurate evaluations. This finding aligns with the experimental results observed in the HDRC database [26]. Furthermore, through additional analysis of the statistical data, we identified a new observation that (2) existing IQA models show varying performance across databases with different data distributions. When using the same perceptual mapping method to assess the IQA models on different databases, certain models exhibit significant performance differences. For example, Gamma-DISTS performs better than Gamma-GMSD on the HDRC and Narwaria2013 datasets, whereas no significant difference is observed on the other three databases. A possible explanation is that the same perceptual mapping method may project different databases into distinct feature spaces compared with their original data distributions. Some IQA algorithms may be better suited to one type of feature space, while others perform more effectively in different feature spaces. From these experiments, we can conclude two limitations of existing HDR IQA models, as follows: (1) the sensitivity of IQA models to the characteristics of different datasets, and (2) the influence of perceptual mapping methods on the consistency of quality evaluations across databases. These findings suggest the need for further optimization and validation of IQA models and perceptual mapping methods for HDR content.

Accurate IQA models are foundational for a wide range of tasks and applications. For instance, in super-resolution, robust IQA models [82] ensure that the enhanced images maintain high visual quality and fidelity, reflecting the original content. In object detection, effective IQA metrics [83] help improve the clarity and precision of detected objects, contributing to more reliable outcomes in complex scenes. Additionally, in image compression, accurate IQA models [84] are critical for evaluating and optimizing compression algorithms to preserve image quality while reducing file size. Beyond these areas, IQA models are integral to various other applications, such as image enhancement, medical imaging, and virtual reality, where they play a crucial role in assessing and improving the quality of visual content. To develop more accurate objective quality models for HDR images, future work can focus on the following aspects: *(1) Pre-processing.* A more perceptually aligned mapping model could be designed to enhance the consistency between the tone-mapped image and the human visual system (HVS). This would ensure that the processed images better reflect human perception. *(2) Feature extraction.* Existing deep learning models can be further optimized for the more efficient extraction of visual features. For instance, transformers [85], specifically vision transformers (ViTs) [86], have shown great potential in capturing global dependencies in images [87,88]. ViTs work by splitting the image into patches, treating each patch as a token, and then applying self-attention mechanisms to model the relationships between different regions of the image. This ability to capture long-range dependencies makes ViTs particularly suitable for IQA, as they can effectively represent subtle perceptual differences. *(3) Learning strategy.* Considering the significance of a subjective visual quality assessment, a promising approach to improve model accuracy could be the adoption of a Human-in-the-Loop strategy [83], which involves incorporating human feedback into the training loop, allowing the model to adjust based on real-time subjective input. This approach enhances the model’s ability to align with human perception and improve generalization to unseen data. To sum up, advancing HDR IQA models requires attention to pre-processing, feature extraction, and innovative training strategies. By focusing on these areas, we can build models that have a higher consistency with the HVS, thereby providing more accurate quality assessment methods for a broader range of image processing tasks.

## Figures and Tables

**Figure 1 jimaging-10-00243-f001:**
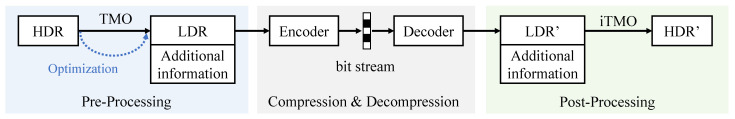
Overview of the HDR image compression schemes.

**Figure 2 jimaging-10-00243-f002:**
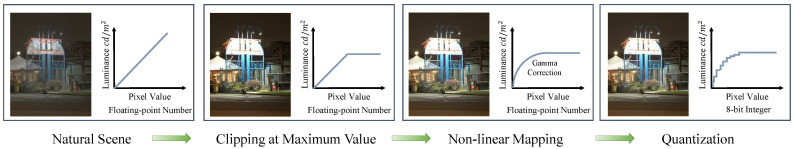
The production process of the LDR image in the sensor.

**Figure 3 jimaging-10-00243-f003:**
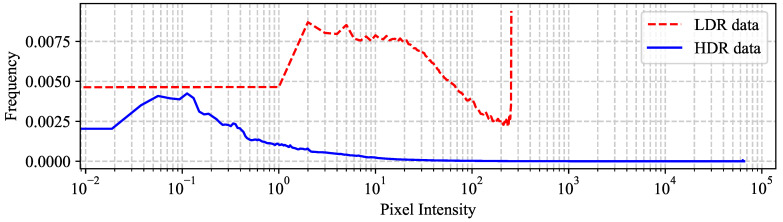
Average histogram of the HDR and LDR images in the SIHDR [52] database.

**Figure 5 jimaging-10-00243-f005:**
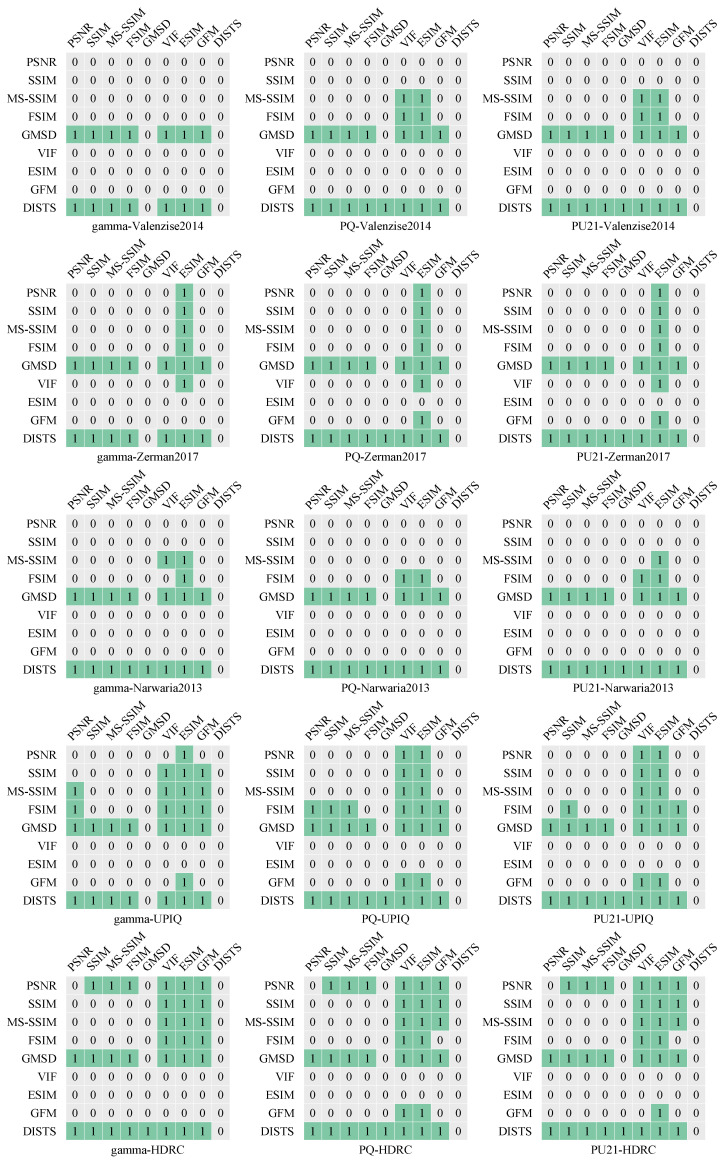
Statistical analysis using the left-tail F-test. Cells highlighted in green (with a value of 1) indicate that the model corresponding to the row significantly outperforms the model corresponding to the column, while a value of 0 indicates no significant difference.

## Data Availability

The original contributions presented in the study are included in the article, further inquiries can be directed to the authors.

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
