# Peer review of "Overview of High-Dynamic-Range Image Quality Assessment"

_2313-433X, 2024, doi:10.3390/jimaging10100243_

Round 1

Reviewer 1 Report

Comments and Suggestions for Authors

The manuscript is good however, it needs some improvement. Here are some suggestions, 

The abstract needs the overall finding; the current form is too general. 

The contribution list should include the findings in addition to the promises.

Well-known concepts should be given with more intricate information, like transformer architectures; not in a story-telling style. The recent works should be included in that section to boost the wide range of references.

Reference-based comparisons of numerical findings and discussion are missing. @ I recently reviewed another article from the same field which is titled as "CN-BSRIQA: Cascaded network-blind super-resolution image quality assessment". I am unable to see that in this paper. I would like the reviewers to add recently published articles related to the field and discuss how these newly proposed techniques are playing role in order to enhance the HR images. 

Clarify Methodological Details: Some methodological details could be further elaborated to enhance clarity and reproducibility. Including step-by-step descriptions and explanations of your processes would benefit the reader.

Discuss Limitations: Provide a more detailed discussion of the limitations and potential challenges of your proposed approach. This would offer a balanced perspective and help readers understand the context of your findings.

Comments on the Quality of English Language

Thats fine

Reviewer 2 Report

Comments and Suggestions for Authors

The work gives a thorough description of methods involved in High Dynamic Range (HDR) Image Quality Assessment (IQA). The authors explore both subjective and objective approaches of HDR IQA. The article emphasizes the limitations of current approaches, such as the absence of large subjective databases and the inefficiency of applying Low Dynamic Range (LDR) IQA metrics to HDR images. It also examines existing HDR IQA algorithms, such as those based on HVS modeling, PU encoding, and adaptations of existing IQA measures. 

The presentation is quite straightforward; however, it reads as a concise report on available methods without going into great detail. Improving the document's organization and flow by reorganizing portions may result in more interesting content. However, this is not a criticism, but rather a minor recommendation that may not be within the scope of this work, as extending into these methods is a lengthy pursuit.

I wish the best to the authors.

Reviewer 3 Report

Comments and Suggestions for Authors

The paper presents a comprehensive survey of current methods for High Dynamic Range (HDR) Image Quality Assessment (IQA), a field that has gained significant relevance due to HDR's expanding applications in areas such as security, multimedia, and biomedical imaging. The authors effectively underscore the key challenge in this domain—the difficulty in accurately assessing the quality of HDR images given their wide dynamic range and detailed content.

One of the paper's strengths is its broad coverage of the topic. The survey methodically outlines both subjective and objective approaches to HDR IQA, providing an insightful look into the current landscape of assessment techniques. The authors make a compelling case for the necessity of improved IQA metrics, which is crucial given HDR's ability to deliver a more authentic visual experience compared to traditional imaging.

However, while the paper succeeds in presenting a detailed overview of HDR IQA, its depth could have been improved by including more concrete examples or case studies that demonstrate the real-world application of these assessment methods. The literature review on human in the loop in visual data should be discussed, as follows "Empowering Zero-Shot Object Detection: A Human-in-the-Loop Strategy for Unveiling Unseen Realms in Visual Data. In: Duffy, V.G. (eds) Digital Human Modeling and Applications in Health, Safety, Ergonomics and Risk Management. HCII 2024. Lecture Notes in Computer Science, vol 14711. Springer, Cham." Such inclusions could make the survey more practical for researchers looking to apply these methodologies to specific domains.

The paper also acknowledges the limitations of existing approaches and identifies several areas for future research, which is commendable. The discussions on the challenges and opportunities for improvement provide a solid foundation for future investigations in this field. Nevertheless, a more focused critique of the specific weaknesses in current IQA metrics might have been useful to guide researchers more precisely.

Overall, this paper serves as a valuable resource for both novice and experienced researchers in the HDR IQA domain. It offers a well-rounded and systematic overview of current techniques while pointing toward potential avenues for innovation. This work will likely contribute to the ongoing development of more efficient and reliable IQA methods for HDR images. Thus, I recommend accepting the paper after minor changes.

Comments on the Quality of English Language

The paper presents a comprehensive survey of current methods for High Dynamic Range (HDR) Image Quality Assessment (IQA), a field that has gained significant relevance due to HDR's expanding applications in areas such as security, multimedia, and biomedical imaging. The authors effectively underscore the key challenge in this domain—the difficulty in accurately assessing the quality of HDR images given their wide dynamic range and detailed content.

One of the paper's strengths is its broad coverage of the topic. The survey methodically outlines both subjective and objective approaches to HDR IQA, providing an insightful look into the current landscape of assessment techniques. The authors make a compelling case for the necessity of improved IQA metrics, which is crucial given HDR's ability to deliver a more authentic visual experience compared to traditional imaging.

However, while the paper succeeds in presenting a detailed overview of HDR IQA, its depth could have been improved by including more concrete examples or case studies that demonstrate the real-world application of these assessment methods. The literature review on human in the loop in visual data should be discussed, as follows "Empowering Zero-Shot Object Detection: A Human-in-the-Loop Strategy for Unveiling Unseen Realms in Visual Data. In: Duffy, V.G. (eds) Digital Human Modeling and Applications in Health, Safety, Ergonomics and Risk Management. HCII 2024. Lecture Notes in Computer Science, vol 14711. Springer, Cham." Such inclusions could make the survey more practical for researchers looking to apply these methodologies to specific domains.

The paper also acknowledges the limitations of existing approaches and identifies several areas for future research, which is commendable. The discussions on the challenges and opportunities for improvement provide a solid foundation for future investigations in this field. Nevertheless, a more focused critique of the specific weaknesses in current IQA metrics might have been useful to guide researchers more precisely.

Overall, this paper serves as a valuable resource for both novice and experienced researchers in the HDR IQA domain. It offers a well-rounded and systematic overview of current techniques while pointing toward potential avenues for innovation. This work will likely contribute to the ongoing development of more efficient and reliable IQA methods for HDR images. Thus, I recommend accepting the paper after minor changes.

Reviewer 4 Report

Comments and Suggestions for Authors

IN this manuscript authors discussed about high dynamic range for image quality assessment. The manuscript suffers in the following aspects. The abstract should have mentioned the measures and the databases. 

The organization needs to be improved. Clarity is missing.  

Comments on the Quality of English Language

Needs through checking. 

Round 2

Reviewer 1 Report

Comments and Suggestions for Authors

My concerns are addressed. 

Reviewer 4 Report

Comments and Suggestions for Authors

Well Organized and can be considered for publication. 

Comments on the Quality of English Language

NA